# Grammar-Constrained Decoding for Structured NLP Tasks without Finetuning

**Saibo Geng,**[◇] **Martin Josifoski,**[◇] **Maxime Peyrard,**[*♣] **Robert West**[◇]

[◇]EPFL ♣Université Grenoble Alpes, CNRS, Grenoble INP, LIG

{saibo.geng, martin.josifoski, robert.west}@epfl.ch, maxime.peyrard@univ-grenoble-alpes.fr

## Abstract

Despite their impressive performance, large language models (LMs) still struggle with reliably generating complex output structures when not finetuned to follow the required output format exactly. To address this issue, *grammar-constrained decoding* (GCD) can be used to control the generation of LMs, guaranteeing that the output follows a given structure. Most existing GCD methods are, however, limited to specific tasks, such as parsing or code generation. In this work, we demonstrate that formal grammars can describe the output space for a much wider range of tasks and argue that GCD can serve as a unified framework for structured NLP tasks in general. For increased flexibility, we introduce *input-dependent grammars*, which allow the grammar to depend on the input and thus enable the generation of different output structures for different inputs. We then empirically demonstrate the power and flexibility of GCD-enhanced LMs on (1) information extraction, (2) entity disambiguation, and (3) constituency parsing. Our results indicate that grammar-constrained LMs substantially outperform unconstrained LMs or even beat task-specific finetuned models. Grammar constraints thus hold great promise for harnessing off-the-shelf LMs for a wide range of structured NLP tasks, especially where training data is scarce or finetuning is expensive. Code and data: https://github.com/epfl-dlab/GCD.

## 1 Introduction

Pretrained language models (LMs) have achieved impressive results across a range of tasks, such as machine translation, summarization, and dialogue generation (Brown et al., 2020; Touvron et al., 2023). All of these models are pretrained on next-token prediction task, encouraging researchers to cast other NLP tasks in the same autoregressive generation framework. By framing tasks as autoregressive generation, pretrained language models

---
*Work done while at EPFL.

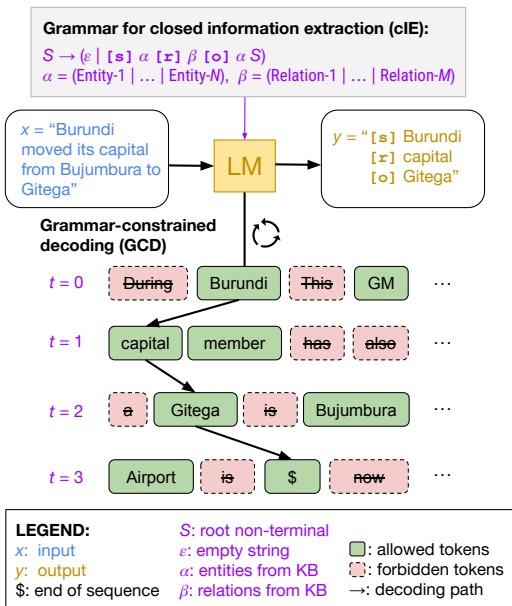

Figure 1: **Grammar-constrained decoding (GCD),** applied to the task of closed information extraction, where the goal is to extract a list *y* of subject–relation–object triplets from the input text *x*. Subjects and objects are constrained to be Wikidata entities, relations to be a Wikidata relation. During decoding, only valid token continuations compliant with the grammar are considered. For simplicity, we omit the special marker symbols [s], [r], and [o] in the schema of the generation process.

can be finetuned for specific tasks with minimal modifications to the training process while still benefiting from the advantages of pretraining. More recently, the scaling of language models to larger sizes has introduced notable in-context learning capabilities (Brown et al., 2020; Radford et al., 2019; Schick and Schütze, 2021), such that large language models (LLMs) can quickly and effectively adapt to new tasks even without finetuning, when shown only few demonstrations as part of their context.

Certain important tasks, however, such as closed information extraction (cIE), entity disambiguation (ED), or constituency parsing (CP), require the output to follow a predefined format and adhere to

a restricted vocabulary (entities, relations, senses, dependency labels, etc.). Whereas LMs excel at generating free-form text, they are not specifically designed for structured prediction tasks where only a small subset of the output space is valid. Consequently, structured prediction tasks present unique challenges because constrained output spaces demand both structural coherence and compliance with a predefined vocabulary. For instance, Josifoski et al. (2023) have demonstrated that few-shot-prompted large language models such as GPT-3.5 struggle with tasks such as cIE. This difficulty is primarily due to the extensive output vocabulary, which includes 2.7 million Wikidata entity names and approximately 1,000 Wikidata relation names, which is too vast to be conveyed with just a few demonstration examples.

One way forward is to finetune LMs for specific tasks, which involves linearizing the desired output format into a string format, thus enabling training through standard next-token prediction. For instance, De Cao et al. (2021) and Josifoski et al. (2022) successfully applied this technique to ED and cIE, respectively. However, this approach has limitations: it necessitates an expensive finetuning pipeline for each new task, which lacks flexibility and requires bespoke training data. Orthogonally, constrained decoding (Tromble and Eisner, 2006) is a technique that can be used to enforce constraints on the output space of an autoregressive language model during inference. Constrained decoding has been used in semantic role labeling (Deutsch et al., 2019), constituency parsing (Deutsch et al., 2019), code generation (Scholak et al., 2021), and entity disambiguation (De Cao et al., 2021). The constraints have been expressed in the form of finite-state automata (Deutsch et al., 2019) or trie-based data structures for fast lookup (De Cao et al., 2021).

In this work, we show that, for a much wider range of NLP tasks, the respective output space can be described with a formal grammar, giving rise to a unified framework for structured NLP tasks. Given an appropriately defined grammar, we use an incremental parser to play the role of a completion engine, which determines the set of valid next tokens given the current prefix. We integrate this completion engine with a pretrained language model to iteratively generate sequences that are *valid* according to the grammar and *plausible* according to the LM (cf. Fig. 1). From a practical perspective, the *grammar-constrained decoding* (GCD) framework allows researchers to focus on writing the grammar while ignoring the implementation details of the constrained decoding process. This is in contrast to previous work, where the constraints were expressed in the form of finite-state automata or trie-based data structures, which require a significant engineering effort to implement. We envision GCD to be as simple to use as regular expressions, in the sense that the user can specify a desired output structure in a declarative way, and the LM-generated sequences will be guaranteed to be valid. With the introduction of *input-dependent grammars,* the scope of tasks that can be tackled with GCD can be further extended to tasks such as entity disambiguation and entity linking, where the output space is not fixed but depends on the input. We show that, by combining GCD with powerful LLMs, we achieve remarkable improvements with few-shot learning, even rivaling the performance of finetuned task-specific models. This is particularly exciting because it shows that LMs can be used to solve a much wider range of structured tasks than before, without the need for finetuning.

Our contributions can be summarized as follows:

1. We demonstrate that the output spaces of many structured NLP tasks can be formulated as formal languages, thus converting the tasks into grammar-constrained decoding problems. This formulation provides a unified framework to tackle structured NLP tasks.
2. We introduce input-dependent grammars, which extend the set of tasks that can be tackled with GCD. We show that this can be useful, among others, for tasks such as ED and CP.
3. Through empirical experiments, we demonstrate the effectiveness of GCD on three structured NLP tasks: cIE, ED, and CP. We show that our method can achieve competitive results on cIE and ED without any finetuning.

## 2 Method

We now describe GCD and how it can constrain the output of LMs at decoding time based on a formal grammar. We first explain how to specify the output spaces of various NLP tasks via formal grammars. Then we show how an incremental parser can be used as a completion engine to constrain the LM's generation process to produce grammatically valid outputs only.

**(1) Closed information extraction:** see Fig. 1

**(2)\* Entity disambiguation:** $S \to \ell\, m\, [\alpha]\, r$, where $\ell$ is left context of mention $m$, $r$ is right context, and $\alpha$ is disjunction of candidate entities for mention $m$

**(3)\* Constituency parsing:** $S \to B_{0,0}$; $B_{i,j} \to [\alpha\ (B_{i,j+1} \mid C_{i,j+1})$; $C_{i,j} \to x_i\ (C_{i+1,j} \mid E_{i+1,j})$; $C_{n,j} \to E_{n,j}$; $E_{i,j+1} \to ]\,(E_{i,j} \mid B_{i,j})$; $E_{n,j+1} \to ]\,E_{n,j}$; $E_{n,0} \to \varepsilon$, where $\alpha = (S \mid NP \mid VP \mid \ldots)$

**(4)\* Coreference resolution:** $S_i \to x_i\ [(x_1 \mid \ldots \mid x_n \mid \perp)]\ S_{i+1}$; $S_n \to \varepsilon$, where $\perp$ means "no referent"

**(5)\* Part-of-speech tagging:** $S_i \to x_i\ [(NOUN \mid VERB \mid ADJ \mid \ldots)]\ S_{i+1}$; $S_n \to \varepsilon$

**(6)\* Dependency parsing:** $S_i \to x_i\ [(ROOT \mid NSUBJ \mid DOBJ \mid \ldots)\ (x_1 \mid \ldots \mid x_n \mid \perp)]\,S_{i+1}$; $S_n \to \varepsilon$, where $\perp$ means "no head"

**(7)\* Word sense disambiguation:** $S_i \to x_i\ [\alpha_i]\ S_{i+1}$; $S_n \to \varepsilon$, where $\alpha_i$ is the disjunction of all WordNet glosses of word $x_i$

**(8)\* Phrase chunking:** $S \to B_0$; $B_i \to [C$; $B_n \to \varepsilon$; $C_i \to x_i\ (C_{i+1} \mid \alpha)\ B_{i+1}$; $C_n \to \alpha]$, where $\alpha = (NP \mid VP \mid PP \mid \ldots)$

**(9)\* Semantic role labeling:** Same as phrase chunking, but with $\alpha = (TARGET \mid ARG0 \mid ARG1 \mid \ldots)$

**(10)\* Entity linking:** Same as phrase chunking, but with $\alpha$ the disjunction of all KB entity names (or $\perp$ for "no entity")

**(11)\* CCG parsing:** Same as constituency parsing, but with syntactic types (e.g., $(S \backslash NP)/NP$) instead of constituent labels. Extra constraints ensure that nodes have at most two children and that syntactic types combine correctly.

**(12)\* Question answering:** $S \to [q]\ [A]$; $A \to (\varepsilon \mid \alpha\, A)$, where $q$ is the question and $\alpha$ the disjunction of all vocabulary words

**(13)\* Extractive summarization:** $S \to (\varepsilon \mid [\alpha]\, S)$, where $\alpha$ is the disjunction of all sentences from input $x$

**(14)\* Semantic parsing with $\lambda$-calculus:** A logical form is a rooted tree, generated by a context-free grammar

Figure 2: **Formal grammars for 14 structured NLP tasks,** highlighting the general applicability of grammar-constrained decoding. All 14 grammars are context-free (mostly regular). \* marks input-dependent grammars. Inputs $x = \langle x_0, \ldots, x_{n-1} \rangle$ are sequences of lexical units (e.g., words); $0 \le i \le n-1$; single capital letters are non-terminal symbols; $S$ or $S_0$ is the start symbol; $\varepsilon$ is the empty string; $[$ and $]$ are special terminal symbols.

## 2.1 NLP tasks as formal languages

The input $x$ and output $y$ of NLP tasks are typically sequences of tokens, $x = \langle x_0, \ldots, x_{n-1} \rangle$ and $y = \langle y_0, \ldots, y_{m-1} \rangle$. Whereas the input $x$ is usually arbitrary, for many tasks the output $y$ needs to follow a specific structure. For instance, in information extraction, $y$ is required to consist of subject–relation–object triplets (cf. Fig. 1). Since formal languages provide a rigorous and complete framework for describing the structure of any computable set of object (according to the Church–Turing thesis), they offer a promising way to define the output spaces of structured NLP tasks. In order to define the formal languages corresponding to the output spaces of structured NLP tasks, our framework relies on formal grammars, a universal formalism that can describe any formal language. For tractability, we focus on the class of context-free grammars.

**Token-level formal grammars.** A formal grammar $G$ is defined as a tuple $(V, \Sigma, P, S)$ where

- $V$ is a finite set of non-terminal symbols,
- $\Sigma$ is a finite set of terminal symbols,
- $P$ is a finite set of production rules,
- $S \in V$ is the start symbol.

We illustrate the suitability of formal grammars for specifying the output spaces of structured NLP tasks in Fig. 2 using 14 common tasks as examples.

In our approach, the user first writes a formal grammar $G$ over characters. In order to obtain a token-level grammar $G_{\text{tok}}$ that can be used to constrain the direct output of LMs—token sequences—we use the token set $\Sigma_{\text{tok}}$ as terminal symbols and apply the tokenizer to the sequences of terminal symbols appearing in the rules $P$, obtaining the token-level rules $P_{\text{tok}}$. This yields the token-level grammar $G_{\text{tok}} = (V, \Sigma_{\text{tok}}, P_{\text{tok}}, S)$, which describes the same language as the character-level grammar $G$. We then use an incremental parser (see below) to decide whether a token sequence $y$ is in the language generated by $G_{\text{tok}}$.

Despite its simplicity, this approach has some limitations. Widely used tokenization methods such as BPE (Sennrich et al., 2016) allow the same string to be tokenized in different ways. For example, the string "[[[" can be tokenized as "[[ [", "[[[" or "[ [ [", all of which would be detokenized to the original string "[[[". To avoid this ambiguity, we can add extra spaces between the brackets in the grammar and force the single bracket to be a token. This approach is, however, not principled and relies on a specific tokenizer. We therefore believe that a principled approach for defining token-level grammars is an interesting direction for future work.

**Grammatical Framework.** Since the token-level grammar $G_{\text{tok}}$ is tokenizer-dependent, there are in general multiple token-level grammars for the same grammar $G$. This one-to-many mapping from a character-level grammar to a token-level grammar is analogous to the one-to-many mapping from abstract syntax trees to different programming languages. For this reason, we adopt Grammatical Framework (GF) (Ranta, 2019) to define both the grammar $G$ and the token-level grammar $G_{\text{tok}}$. GF is a meta-language for multilingual grammar applications, which allows us to define an abstract grammar as well as concrete grammars for linearizing

abstract syntax trees into different tokenizer-specific "languages". In our case, the abstract grammar is the character-level grammar $G$ and the concrete grammars are the token-level grammars for different tokenizers.

**Input-dependent grammars (IDG).** While the output of many NLP tasks can be described by a formal grammar, some tasks require a grammar that is dependent on the input sequence. For example, in entity dismbiguation, the output needs to be constrained to the set of entity candidates, which usually contains dozens of entity names semantically related to the target mention in the input sequence. In constituency parsing, outputs are parse trees whose terminal nodes are the input tokens. These two tasks both require a grammar that is dependent on the input. Existing work has focused on using a single grammar to constrain the decoding regardless of the input sequence. Whereas this is suitable for tasks where the output space is independent of the input sequence, such as code generation (Poesia et al., 2022) or information extraction (Josifoski et al., 2022) (cf. Fig. 1), 13 of the 14 tasks listed in Fig. 2 (those with an asterisk) require an input-dependent grammar.

## 2.2 Grammar-constrained decoding (GCD)

The LM decoding process produces tokens one by one. To enforce the formal grammar, we intervene during decoding by pruning the probability distribution to include only the subset of tokens that are allowed by the formal grammar. The subset of allowed tokens is returned by an incremental parser, which takes the partially generated sequence and the formal grammar as inputs and returns the set of next allowed tokens. The role of the parser can be abstracted as a *completion engine* (Poesia et al., 2022), which derives completion candidates from the partial sequence and the formal grammar $G$. In this work, we use the *incremental parser* of Grammatical Framework (Angelov, 2009) as the completion engine. This process is compatible with any decoding algorithm, including greedy decoding, beam search, top-$k$ sampling, etc. GCD can also be applied to any autoregressive language model, provided that we have access to the distribution over the vocabulary at each decoding step. Since API-based services such as OpenAI do not provide access to the distribution over the vocabulary, they cannot be used with GCD.

## 2.3 Few-shot learning with GCD

To adapt a pretrained LM to a new task, one can either finetune the LM on task-specific training data or use few-shot learning if the LM is powerful enough. In this work, we use GCD in conjunction with few-shot learning to adapt a pretrained large LM (LLM) to new tasks. Instead of prompting the LLM to generate free-form text, we use GCD to constrain the output to be grammatically correct. This allows us to leverage the LLM's few-shot learning capability together with the grammar-induced knowledge of the task's output structure.

For the same task, we can use different grammars to constrain the output of the LLM. For example, in entity disambiguation, we can use a grammar $G_1$ that depends on the input sequence to constrain the output to the input-specific candidate set, or we could use a grammar $G_2$ that is independent of the input to constrain the output to be any valid entity name. While both $G_1$ and $G_2$ can be used to constrain the output of the LLM, $G_1$ reduces the search space more and thus is more effective.

## 3 Experimental setup

Although GCD can be applied to many tasks, we concentrate on three tasks in order to showcase its effectiveness: closed information extraction (cIE), entity disambiguation (ED), and constituency parsing (CP). The first two tasks are examples where the output is restricted to a predefined set of entities and relations, while the third is an example of a task where the output is a complex tree structure. All three tasks are challenging for LLMs in the few-shot setting, and we show that GCD can significantly improve LLM performance.

### 3.1 Closed information extraction (cIE)

**Task description.** The goal of closed information extraction (cIE) is to extract a comprehensive set of facts from natural-language text. Formally, given a knowledge base (KB) containing a collection of entities $\mathscr{E}$ and a collection of relations $\mathscr{R}$, the goal is to extract the complete set $y_{\text{set}} \subset \mathscr{E} \times \mathscr{R} \times \mathscr{E}$ of fact triplets from a given input text $x$.

**Grammar.** We implement the grammar shown in Fig. 1. Outputs are sets $y_{\text{set}}$ of triplets represented as structured sequences $y$ of tokens. Each triplet consists of a subject entity name, a relation name, and an object entity name, each preceded by the special marker [s], [r], or [o], respectively. For

instance, the two-triplet set $y_{\text{set}} = \{$(Witchita, cast member, John Smith); (Witchita, instance of, film)$\}$ is mapped to $y =$ "[s] Witchita [r] cast member [o] John Smith [s] Witchita [r] instance of [o] film". Entity and relation names are restricted to a predefined set of entities (2.7M) and relations (888) from the Wikidata KB (Vrandečić, 2012). The grammar is context-free and allows an arbitrary number of triplets to be generated, including zero.

**Dataset and evaluation metric.** We use the SynthIE-text dataset (Josifoski et al., 2023), a synthetic dataset generated by prompting GPT-3.5. This dataset, in comparison to previous ones like REBEL (Huguet Cabot and Navigli, 2021), is characterized by its larger size, increased diversity, and higher quality according to human ratings (Josifoski et al., 2023). The SynthIE-text dataset comprises 10K validation samples and 50K test samples. For the purpose of evaluating our method in the few-shot scenario, we exclusively employ the test data. We measure performance via triplet-based micro-precision, recall, and F1-score, following Josifoski et al. (2022).

### 3.2 Entity disambiguation (ED)

**Task description.** Entity disambiguation (ED) is the task of identifying the exact entity from a predefined knowledge base (e.g., Wikidata) referred to by a mention demarcated by special tokens in an input text. In certain cases, the input may also contain a set of candidate entities to narrow the search scope.

**Grammar.** We use grammar 2 of Fig. 2. Following De Cao et al. (2021), the output structure consists of the mention followed by the inferred entity name in square brackets. For instance, given the input "There are two types of electricity: <ent> DC </ent> and AC", the output is represented as "There are two types of electricity: <ent> DC [Direct current] </ent> and AC". The grammar is regular and input-dependent. It forces the model to generate the mention first, followed by an opening square bracket, an entity name from the candidate set, and finally a closing square bracket. The candidate set is mention-dependent and is provided in the dataset. To demonstrate the benefits of using an input-dependent grammar (IDG), we also experiment with an input-independent grammar (IIG). For such a grammar, the candidate set needs to be the entire entity catalog of all entities (e.g., 470K

in the data of Le and Titov (2018)). The constraints imposed by IIG are thus weaker than those of IDG. Moreover, forcing the model via the IDG to repeat the left context (e.g., "There are two types of electricity:") may guide the model (via conditioning) in generating the correct entity name.

**Dataset and evaluation metric.** For the ED task, we employ six widely used datasets: AIDA-CoNLL (Hoffart et al., 2011), MSNBC, ACE2004, AQUAINT, CLUEWEB, and WIKI (Gabrilovich et al., 2013; Guo and Barbosa, 2017). We use only the test data to evaluate the effectiveness of our method in a few-shot learning setting. To measure the performance of our approach, we employ micro-accuracy as the evaluation metric. Further details about the datasets and evaluation metric are provided in Appendix D.

### 3.3 Constituency parsing (CP)

**Task description.** Constituency parsing (CP) is the task of parsing a sentence into a constituency parse tree capturing the syntactic structure of the sentence.

**Grammar.** The output in CP must be a *valid*—but not necessarily correct—constituency parse tree in Penn Treebank format (Sekine and Collins, 2008). A valid parse tree is defined as a tree that satisfies the constraints of *completeness* (every word in the sentence is included somewhere in the parse tree), *balanced brackets* (every right bracket closes a previously unclosed left bracket, and every left bracket is eventually closed by a right bracket), and *label consistency* (the label of terminal and non-terminal nodes is consistent with the Penn Treebank format).

To capture these constraints, we use grammar 3 of Fig. 2. The grammar reproduces the input, represented as a sequence $x = \langle x_0, \ldots, x_{n-1} \rangle$ of words, in left-to-right order, interspersing it with node labels and balanced brackets. In order to guarantee balanced brackets, the non-terminals $B_{i,j}$ count the number of opened left brackets [ using the second subscript index $j$, and the rules ensure that the number of closed brackets can never exceed the number of previously opened brackets. As an example, for the input $x =$ "Nkurunziza leads Burundi from Gitega", one valid parse tree would be $y =$ "[S [NP Nkurunziza][VP leads [NP Burundi][PP from [NP Gitega]]]]".

Note that the grammar in this task needs to be input-dependent due to the aforementioned com-

pleteness constraint. To demonstrate this, we also experiment with an input-independent grammar, a context-free grammar that recursively generates a parse tree of arbitrary size whose terminal nodes are anonymized as XX. This grammar satisfies the balanced-brackets and label-consistency constraints, but not the completeness constraint. As the grammar is context-free, it can generate a parse tree with an arbitrary number of nodes, possibly larger or smaller than the number of words in the input, which would result in an invalid parse tree.

**Dataset and evaluation metric.** We use the test split of Penn Treebank to evaluate the effectiveness of our method in a few-shot learning setting. Since we observed that the LLaMA models used in our experiments struggle to generate fully correct parse trees for long input sentences, both with and without constraints, we use only sentences with gold parse trees shorter than 64 tokens. We report the bracketing F1-score returned by the PYEVALB tool as our main evaluation metric. As we observed that LLaMA without constraints often generates invalid parse trees, we also report *validity* (the percentage of valid parse trees) as an additional metric.

### 3.4  LLMs and prompting

We utilize LLaMA (Touvron et al., 2023) and Vicuna (Chiang et al., 2023) as backbone LMs, without performing any finetuning on downstream tasks. Concretely, we evaluate the LLaMA-{7B, 13B, 33B} and Vicuna-{7B, 13B} models. To construct the prompt, we begin by randomly selecting several data points from the training set and use them to manually craft multiple prompts for each task. For more details about the used prompts and the decoding settings, see Appendices E and F.

### 4  Experimental results

Next, we present the results for each task, showing that, whereas the unconstrained LLaMA and Vicuna models perform poorly, the grammar-constrained versions perform significantly better. We also show input-dependent grammars to be crucial for performance, as they allow the models to adapt to the input and generate more accurate outputs. Out of the tested few-shot-prompted models, LLaMA-33B with input-dependent grammars achieves the best performance on all tasks, even rivaling finetuned models on cIE and ED.

| Method | Precision | Recall | F1 |
|---|---|---|---|
| *Weakly supervised* | | | |
| GenIE T5-base | **49.6** $\pm$ 0.3 | 26.8 $\pm$ 0.2 | 34.8 $\pm$ 0.2 |
| *Few-shot unconstrained* | | | |
| LLaMA-7B | 10.2 $\pm$ 0.5 | 14.3 $\pm$ 0.7 | 11.9 $\pm$ 0.5 |
| LLaMA-13B | 10.3 $\pm$ 0.6 | 17.0 $\pm$ 0.9 | 12.9 $\pm$ 0.6 |
| LLaMA-33B | 14.1 $\pm$ 1.0 | 23.1 $\pm$ 1.4 | 17.5 $\pm$ 1.0 |
| Vicuna-7B | 12.5 $\pm$ 0.2 | 16.7 $\pm$ 0.1 | 14.3 $\pm$ 0.2 |
| Vicuna-13B | 13.4 $\pm$ 0.2 | 15.2 $\pm$ 0.2 | 14.4 $\pm$ 0.2 |
| *Few-shot constrained* | | | |
| LLaMA-7B | 27.9 $\pm$ 0.6 | 20.2 $\pm$ 0.5 | 23.5 $\pm$ 0.5 |
| LLaMA-13B | 36.2 $\pm$ 0.7 | 26.5 $\pm$ 0.5 | 30.6 $\pm$ 0.5 |
| LLaMA-33B | 39.3 $\pm$ 0.9 | **33.2** $\pm$ 0.8 | **36.0** $\pm$ 0.7 |
| Vicuna-7B | 25.4 $\pm$ 0.5 | 15.8 $\pm$ 0.3 | 19.5 $\pm$ 0.3 |
| Vicuna-13B | 38.7 $\pm$ 1.0 | 19.8 $\pm$ 0.8 | 26.1 $\pm$ 0.8 |

Table 1: **Main results for closed information extraction (4 shots),** in terms of precision, recall, and F1-score (micro-averaged, with 90% confidence intervals) on the SynthIE-text-small dataset (Josifoski et al., 2023). Best results in bold. We report the GenIE model (Josifoski et al., 2022) for the supervised setting.

### 4.1  Closed information extraction (cIE)

Results for cIE are reported in Table 1. Unconstrained LLaMA, even with few-shot demonstrations, performs poorly on cIE. This is not surprising, since the cIE task requires generating valid entity and relation names from a knowledge base (Wikidata in our case). Although LLMs have been exposed to Wikidata to a certain extent during pretraining, they still struggle with generating accurate entity and relation names contained in the KB. This can be seen as a special case of the hallucination problem, where the model generates entities and relations not present in the KB.

We see a significant improvement when constraining the generation to only produce valid entities and relations. Notably, LLaMA-33B beats GenIE T5-base (Josifoski et al., 2022), a state-of-the-art autoregressive model specifically trained for the cIE task on supervised data from the REBEL dataset (Huguet Cabot and Navigli, 2021). In the table, we refer to GenIE as weakly supervised because it was not trained on the train split of SynthIE-text, but on REBEL. We observe that the grammar-constrained LLaMA models balance precision vs. recall better than GenIE, achieving a higher F1-score. While GenIE exhibits higher precision, its recall is lower, implying that it misses many entities and relations. This may be because GenIE was optimized for the REBEL dataset and may have memorized the entities and relations in the dataset. As domain-specific training data is often scarce (Dunn et al.,

| Method | AIDA | MSNBC | AQUAINT | ACE2004 | CWeb | WIKI | Avg. |
|---|---|---|---|---|---|---|---|
| *Supervised* | | | | | | | |
| Le and Titov (2018) | 89.6 | 92.2 | 90.7 | 88.1 | 78.2 | 81.7 | 86.8 |
| BLINK w/o candidate set | 79.6 | 80.0 | 80.3 | 82.5 | 64.2 | 75.5 | 77.0 |
| BLINK (Wu et al., 2020) | 86.7 | 90.3 | 88.9 | 88.7 | **82.6** | 86.1 | 87.2 |
| GENRE only AIDA data | 88.6 | 88.1 | 77.1 | 82.3 | 71.9 | 71.7 | 80.0 |
| GENRE (De Cao et al., 2021) | 93.3 | **94.3** | 89.9 | 90.1 | 77.3 | 87.4 | 88.8 |
| ReFinED w/o pretraining | 88.2 | 92.3 | 86.8 | 90.6 | 75.1 | 74.5 | 84.6 |
| ReFinED (Ayoola et al., 2022) | **93.9** | 94.1 | **90.8** | **90.8** | 79.4 | **87.4** | **89.4** |
| *Few-shot unconstrained* | | | | | | | |
| LLaMA-7B | 42.0 | 44.6 | 30.2 | 43.8 | 35.8 | 27.7 | 37.4 |
| LLaMA-13B | 48.1 | 50.2 | 36.2 | 47.5 | 40.7 | 37.2 | 43.3 |
| LLaMA-33B | 62.6 | 63.0 | 42.9 | 56.3 | 48.1 | 51.4 | 54.1 |
| *Few-shot constrained (IIG)* | | | | | | | |
| LLaMA-7B | 56.3 | 57.3 | 61.6 | 54.6 | 50.5 | 47.0 | 54.5 |
| LLaMA-13B | 51.8 | 57.3 | 53.3 | 50.8 | 48.2 | 39.7 | 50.6 |
| LLaMA-33B | 69.8 | 73.3 | 74.9 | 71.7 | 61.6 | 57.6 | 68.2 |
| *Few-shot constrained (IDG)* | | | | | | | |
| LLaMA-7B | 73.4 | 87.6 | 83.2 | 82.9 | 69.4 | 67.1 | 77.2 |
| LLaMA-13B | 75.8 | 86.6 | 82.4 | 84.2 | 68.1 | 68.1 | 77.5 |
| LLaMA-33B | 81.0 | 88.2 | 86.2 | 85.4 | 70.7 | 70.5 | 80.3 |

Table 2: **Main results for entity disambiguation (4 shots),** in terms of micro-accuracy. Width of 90% confidence intervals is between 0.1 and 0.3 for all results. Best results in bold. IIG stands for "input-independent grammar", IDG for "input-dependent grammar".

2022), this result highlights the potential for LLMs to excel on cIE without finetuning.

## 4.2 Entity disambiguation (ED)

Results for ED are reported in Table 2. Whereas unconstrained LLaMA models perform poorly, GCD (either input-dependent [IDG] or input-independent [IIG]) significantly improves the performance of LLaMA. Although there is still a gap with respect to the state-of-the-art model, GENRE (De Cao et al., 2021), grammar-constrained LLaMA-33B performs better than a version of GENRE trained only on the AIDA dataset (without pretraining on Wikipedia). Considering that many domain-specific information extraction tasks have limited data available (Dunn et al., 2022), the constrained LLaMA models can thus be a good choice for low-resource settings. Among the GCD-powered LLaMA models, we observe that IDG performs better than IIG, highlighting the benefits of using an input-dependent grammar. The latter allows the model to leverage an input-specific candidate set, whereas an input-independent grammar can only use the entire knowledge base as the candidate set. We believe this flexibility is crucial for GCD to achieve good performance on various tasks.

## 4.3 Constituency parsing (CP)

Results for CP are reported in Table 3. In contrast to the previous two tasks, the performance of LLMs—with or without GCD—on constituency parsing is much worse when compared to bespoke methods. This is not surprising, as constituency parsing requires syntactic understanding of the input, as opposed to the other two tasks, which only required semantic understanding. Through error inspection, we found that, although the LLMs are able to generate seemingly reasonable output, their outputs are often syntactically incorrect. (For examples, see Appendix I.)

While overall, LLaMA models perform poorly on CP, the GCD-powered LLaMA models still significantly outperform the unconstrained LLaMA models. Importantly, with an input-dependent grammar, GCD guarantees that the generated output is a *valid* constituency parse tree, which is not the case with an input-independent grammar.

In conclusion, GCD substantially improves the performance of LLMs on constituency parsing, but performance still falls short of the F1-scores achieved by supervised methods (95% and above). We do not, however, rule out the possibility that GCD might produce better results once the underlying LLMs become more powerful.

| Method | F1 | Validity |
|---|---|---|
| ***Bespoke methods*** | | |
| Vinyals et al. (2015a) | 92.1 | 98.5 |
| Dyer et al. (2016) | 93.3 | 100.0 |
| Kitaev and Klein (2018) | 95.6 | 100.0 |
| Zhang et al. (2020) | 95.7 | 100.0 |
| ***Few-shot unconstrained*** | | |
| LLaMA-7B | 28.1 | 54.3 |
| LLaMA-13B | 42.8 | 69.4 |
| LLaMA-33B | 42.9 | 64.2 |
| ***Few-shot constrained (IIG)*** | | |
| LLaMA-7B | 34.7 | 65.9 |
| LLaMA-13B | 45.4 | 80.3 |
| LLaMA-33B | 47.1 | 72.3 |
| ***Few-shot constrained (IDG)*** | | |
| LLaMA-7B | 45.8 | 100.0 |
| LLaMA-13B | 53.4 | 100.0 |
| LLaMA-33B | 54.6 | 100.0 |

Table 3: **Main results for constituency parsing (8 shots),** in terms of bracketing F1-score and parse-tree validity. For few-shot-prompted LLaMA models, test set was restricted to gold parse trees shorter than 64 tokens, as LLaMA models perform poorly on longer sentences. For bespoke methods, entire Penn Treebank test set was used. Width of 90% confidence intervals is between 4.0 and 6.0 for all F1-scores. Best results in bold.

## 4.4 Latency

Incremental parsing imposes an additional overhead on top of pure vanilla decoding. To quantify this overhead, we report the latency of pure decoding and compare it to the added latency due to enforcing grammar constraints in Table 4. Note that GCD operates entirely on the CPU, not on the GPU, so GCD latency is measured on a consumer CPU. As shown in Table 4, the added latency from GCD is negligible for the ED and CP tasks. For cIE, GCD adds a modest additional latency comparable or inferior to the latency of pure decoding, depending on the model used (cf. Appendix H for more details).

## 5 Likelihood misalignment in GCD

In the cIE and CP tasks, regardless of model size, the top generation is consistently an empty string (technically, a string "$" consisting of the end-of-sequence token only) and the second most likely generation and subsequent generations are non-empty output sequences. The issue is not unique to a particular LLM, but emerges consistently. It is also similar to an observation by Stahlberg and Byrne (2019), who found that the most likely out-

| Model/task | Latency |
|---|---|
| Unconstrained LLaMA-7B | 54 |
| Unconstrained LLaMA-33B | 87 |
| Unconstrained LLaMA-65B | 136 |
| GCD overhead: cIE | 69 |
| GCD overhead: ED | 1 |
| GCD overhead: CP | 4 |

Table 4: **Per-token decoding latency** (in milliseconds) for unconstrained decoding (top 3 rows, measured on A100 GPU), compared to overhead due to grammar-constrained decoding (bottom 3 rows, measured on consumer CPU).

put of neural machine translation models is usually an empty string.

We hypothesize that the empty-string issue is caused by a likelihood misalignment between the grammar and the language model. We use the cIE task as an example to illustrate the issue. In the case of cIE, the first generated token must either be the left-bracket token "[" or the end-of-sequence token "$". The latter denotes the situation where no triplets can be extracted from the input. With a beam size $k \geq 2$, both "[" and "$" will be included in the beam at the first step. Assume that the likelihood of "[" is $p$ and the likelihood of "$" is $q$. Since "$" denotes the end of the sequence, this generation is considered as a complete generation with a total likelihood of $q$. In the other beam, the generation continues with "[" as the prefix, but as the generation proceeds, the likelihood of the generation may decrease below $q$.

Since LLMs such as LLaMA are trained to maximize the likelihood of human language, the structure imposed by the grammar may be unnatural to the model, especially when the model is not finetuned on the respective task. In the extreme case, the correct ground-truth output could have a likelihood lower than that of the empty string "$", resulting in the latter being returned as the top generation. This intuition gives rise to a simple fix: penalize the model for generating short strings, e.g., by adjusting the length normalization parameter $\alpha$ in the length-adjusted sentence score $S/m^\alpha$, where $S$ is the unadjusted score for the sentence and $m$ is the number of tokens in the sentence. As shown in Table 5, this fix indeed solves the problem.

We also observed that the empty-string issue can be alleviated by using instruction-tuned models. While, without applying the aforementioned length normalization fix, LLaMA-13B always outputs the

| Length normalization $\alpha$ | 1.0 | 1.5 | 2.0 | 2.5 | 3.0 | 3.5 |
|---|---|---|---|---|---|---|
| Top generation = "$"? | ✓ | ✓ | ✓ | × | × | × |

Table 5: **Length normalization** mitigates the empty-string issue: larger $\alpha$ favors longer sequences and $\alpha = 0$ means no effect (results for LLaMA-13B).

empty string "$" as its top generation on the cIE task, the instruction-tuned version (Vicuna-13B) outputs non-empty output as its top generation only 46% of the time. (The smaller Vicuna-7B, however, still always outputs "$" as its top generation.)

## 6 Related work

**Autoregressive structured prediction in NLP.** It has become popular to use autoregressive generative models for structured prediction tasks, as this fits the training mode and specific strengths of language models (Vinyals et al., 2015b; Athiwaratkun et al., 2020; De Cao et al., 2021; Paolini et al., 2021). For instance, Vinyals et al. (2015b) modeled dependency parsing as a sequence generation problem and leveraged LSTMs to tackle it effectively. De Cao et al. (2021) proposed autoregressive language models to address entity linking, entity disambiguation, and document retrieval tasks.

**Constrained decoding.** For tasks where the output needs to satisfy certain constraints, constrained decoding has been proposed to guide the generation process to produce valid outputs. For instance, Hokamp and Liu (2017); Hu et al. (2019); Post and Vilar (2018) proposed lexically-constrained sequence decoding for generation tasks. Anderson et al. (2017) extended the beam search algorithm by pruning the search space based on constraints. Scholak et al. (2021) leveraged an incremental parsing technique to generate approximately valid SQL queries. De Cao et al. (2021) addressed entity disambiguation with trie-based lexical constraints at decoding time to force outputs to be valid entities. Josifoski et al. (2022) addressed closed information extraction by combining trie-based lexical constraints with state-based constraints to force the output to be valid triplet sequences. Wang et al. (2023) used Earley parser–based constraints to force outputs to lie in a domain-specific language.

**Grammar-constrained decoding.** Deutsch et al. (2019) proposed a general framework for push-down automata–based constraints and applied it to parsing tasks, which is equivalent to CFG-based constraints in terms of expressiveness. Yin and Neubig (2017) proposed a grammar-powered neural architecture for general-purpose code generation. Shin et al. (2021) proposed to use grammar-constraints to solve semantic parsing tasks with GPT-3 and envisioned that a semantic parser can be built by combining a large language model with a carefully designed grammar. Roy et al. (2022) released a toolkit for grammar-constrained decoding to generate valid meaning representations. Finally, Stengel-Eskin et al. (2023) tested the ability of LLMs to solve parsing task under ambiguity and used grammar constraints to ensure the grammaticality of the output.

## 7 Conclusion

This work introduces grammar-constrained decoding (GCD) for enhancing the few-shot performance of LLMs on challenging structured NLP tasks. We showed that many NLP tasks can be formulated as formal grammars, and that GCD can enhance the performance of LLMs on these tasks. With input-dependent grammars, we further broaden the scope of GCD to accommodate tasks where the set of valid output structures is constrained by the given input. Our experiments indicate that, whereas unconstrained LLMs have difficulty tackling tasks requiring structured outputs, GCD can significantly bolster the performance of LLMs on such tasks. We envision GCD as a swift and cost-efficient adaptation strategy, allowing LLMs to generate reliable structured outputs without the necessity for costly and cumbersome finetuning. Given the fast pace of LLM evolution, we anticipate the usefulness of GCD for boosting pretrained LLMs to further increase over time.

**Best practices for GCD.** We conclude with considerations regarding the effective use of GCD.

1. GCD is more effective with larger LLMs. When possible, use the largest available LLM.

2. Grammars should be as restrictive as possible. Consider using input-dependent grammars.

3. While GCD is broadly applicable to many tasks, it is not a silver bullet. Tasks that require syntactic understanding of the input (e.g., constituency parsing) are less suitable for GCD.

4. In the presence of task-specific training data, finetuning a small model may still yield better performance, at the cost of decreased convenience (cf. Appendix A).

## Limitations

**Compatibility with API-based LLMs.** GCD works by modifying the decoding process of an LLM. If the LLM is hosted in the cloud (as is the case for OpenAI's GPT series) and the API does not provide user control over the decoding process, GCD cannot be used.

**Latency.** The introduction of constraints into the generation process increases latency. The extra overhead of GCD is introduced by the completion step where the next allowed token is determined based on the current prefix and the grammar. The speed of the completion step depends on the complexity of the grammar and the parsing algorithm of the underlying completion engine. In case the grammar is simple, we do not observe a significant increase in latency (the overhead is negligible compared to the latency of the LM). However, in case the grammar contains a large number (e.g., millions) of rules, such as the grammar for the cIE task, the latency of the incremental parsing step grows. (See results in Table 4.)

## Acknowledgements

We thank Viktor Kunčak and Chris Wendler for insightful discussions and suggestions, as well as the Grammatical Framework community, especially Inari Listenmaa, for answering our questions. We also thank Zheng Zhou and Yifei Li for their help setting up the infrastructure for the experiments. West's lab is partly supported by grants from Swiss National Science Foundation (200021_-185043), Swiss Data Science Center (P22_08), H2020 (952215), Microsoft Swiss Joint Research Center, and Google, and by generous gifts from Facebook, Google, and Microsoft.

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

## A  Fine-tuning vs. GCD

It is worth considering whether to prefer fine-tuning or GCD to adapt LLMs to structured prediction tasks. Fine-tuning can be applied to either a small model, such as T5 (Raffel et al., 2019), or directly to a large model, like LLaMA (Touvron et al., 2023). There are three crucial factors to take into account:

1. availability of training data,
2. computational cost of fine-tuning, and
3. performance improvement.

Constrained decoding eliminates the need for training data and incurs only the computational cost of running the LM, with small overhead coming from the incremental parser. Fine-tuning a small model is affordable and currently represents the state-of-the-art (SOTA) approach for most structured prediction tasks when training data is available (De Cao et al., 2021; Jimenez Gutierrez et al., 2022). However, it necessitates a substantial amount of data. Although fine-tuning a large model is expensive, it is highly data-efficient (Brown et al., 2020). However, recent advancements in efficient fine-tuning of large models (Hu et al., 2021) have significantly reduced the computational cost and hardware requirements. We view GCD as a rapid and cost-effective adaptation strategy that enables LMs to produce reliable structured outputs without the need for fine-tuning.

## B  Grammatical Framework

Grammatical Framework (GF) is a programming language for multilingual grammar applications. It is a special-purpose language for grammars, like YACC, Bison, Happy, BNFC, but not restricted to programming languages. It is a functional programming language, like Haskell, Lisp, OCaml, SML, Scheme, but specialized to grammar writing. For example, the following is a GF grammar to generate simple English sentences about food:

```
abstract Food = {
  flags startcat = Comment ;
  cat
    Comment ; Item ; Kind ; Quality ;
  fun
    Pred : Item -> Quality -> Comment ;
    This, That : Kind -> Item ;
    Mod : Quality -> Kind -> Kind ;
    Wine, Cheese, Fish : Kind ;
    Very : Quality -> Quality ;
    Fresh, Warm, Italian,
}
```

```
Expensive, Delicious, Boring : Quality ;
```
Listing 1: The abstract syntax of the Food grammar.

### B.1  Abstract Syntax vs. Concrete Syntax

Constrained decoding works by pruning the set of allowed tokens-id at each decoding step. This requires that both the grammar and the completion engine work at the token-id level. Since the tokenization scheme of LMs are usually different from each other, the grammar becomes dependent on the tokenization scheme of the LM. This brings two challenges: (1) the grammar needs to be redefined for each LM (tokenization scheme), and (2) the debugging of the grammar is difficult because the grammar is defined at the token-id level. We propose to decouple the grammar from the tokenization scheme of the LM by defining an abstract grammar and a couple of concrete grammars. The abstract grammar is defined at the text level, which is human-readable and independent of the tokenization scheme of the LM. The concrete grammars are defined at the token-id level, which are dependent on the tokenization scheme of the LM and work with the completion engine. Once an abstract grammar is defined, the concrete grammars can be automatically translated from the abstract grammar. This is similar to the process of compiling a high-level programming language to a low-level assembly language. The separation of the abstract grammar and the concrete grammar is implemented in GF. For example, the predication rule

```
Pred. Comment ::= Item "is" Quality
```
Listing 2: BNF Notation

now becomes two rules:

```
fun Pred : Item -> Quality -> Comment ;
lin Pred item quality = item ++
        "is" ++ quality ;
```
Listing 3: Grammatical Framework Notation

All that matters in a **linearization rule** is that it defines a string as a function of the variables that it depends on. A GF grammar consists of two parts: **abstract syntax** and **concrete syntax**. Below is the concrete syntax of the aforementioned Food grammar:

```
concrete FoodEng of Food = {
lincat
  Comment, Item, Kind, Quality = Str ;
lin
  Pred item quality = item ++ "is"
      ++ quality ;
  This kind = "this" ++ kind ;
```

```
    That kind = "that" ++ kind ;
    Mod quality kind = quality ++ kind ;
    Wine = "wine" ;
    Cheese = "cheese" ;
    Fish = "fish" ;
    Very quality = "very" ++ quality ;
    Fresh = "fresh" ;
    Warm = "warm" ;
    Italian = "Italian" ;
    Expensive = "expensive" ;
    Delicious = "delicious" ;
    Boring = "boring" ;
}
```

Listing 4: The concrete syntax of the Food grammar.

And also the concrete syntax of the Food grammar in Italian:

```
concrete FoodIta of Food = {
    lincat
      Comment, Item, Kind, Quality = Str ;
    lin
    Pred item quality = item ++ "e"
        ++ quality ;
    This kind = "questo" ++ kind ;
    That kind = "quel" ++ kind ;
    Mod quality kind = kind ++ quality ;
    Wine = "vino" ;
    Cheese = "formaggio" ;
    Fish = "pesce" ;
    Very quality = "molto" ++ quality ;
    Fresh = "fresco" ;
    Warm = "caldo" ;
    Italian = "italiano" ;
    Expensive = "caro" ;
    Delicious = "delizioso" ;
    Boring = "noioso" ;
}
```

Listing 5: The concrete syntax of the Food grammar in Italian.

## B.2 Multilingual Grammars

A **multilingual grammar** is a system with **one** abstract syntax and **any number** of concrete syntaxes. While GF was originally designed for machine translation, it happens to be well suited to handle multi-tokenizations. Different large language models have different tokenizers. For the same sentence, their representations in token id space are different. This phenomenon is very similar to the multilingual grammar scenario, where the same abstract syntax tree can be linearized into different languages as shown in in the Food grammar.

## B.3 Expressivity

Grammatical Framework (GF)'s incremental parser, supports PMCFG (Parallel Multiple Context-Free Grammars). PMCFG lies between mildly context-sensitive and fully context-sensitive grammars (Seki et al., 1991).

## C   IE Task Settings

We use the same KB as in (Josifoski et al., 2023), which contains 2.7M entities and 888 relations from the WikiData KG (Vrandečić, 2012). We use SynthIE-text dataset (Josifoski et al., 2023), a synthetic dataset for the IE task generated from prompting GPT3.5 model. This dataset consists of 10K validation and 50K test samples. It was shown to have a better quality than the widely used REBEL dataset (Huguet Cabot and Navigli, 2021).

## D   ED Task Settings

**Dataset Preprocessing.** Typically, solving the ED task requires contextual information. However, through our observations, we have noticed that the performance of Language Models (LMs) tends to degrade when exposed to long contexts. To mitigate this issue, we have limited the left and right context surrounding the mention to only 10 tokens each.

**Out of Knowledge Base Mention.** It's important to note that some of these datasets contain mentions that are not present in the knowledge base (YAGO_KB) To ensure consistency and accuracy, we filter out these mentions, exclusively utilizing the ones that are available. We report the number of data points whose target entity is not in the knowledge base. These data points are filtered out in the experiments with input-independent grammar.

- AIDA-CoNLL: 0 out of 4485

- ACE2004: 0 out of 257

- AQUAINT: 4 out of 727

- ClueWeb: 12 out of 11154

- MSNBC: 0 out of 656

- Wikipedia: 23 out of 6821

In the experiments with input-dependent grammar, we also filter out the data points whose candidate set is empty.

- AIDA-CoNLL: 0 out of 4485

- ACE2004: 17 out of 257

- AQUAINT: 24 out of 727

- ClueWeb: 44 out of 11154

- MSNBC: 5 out of 656

- Wikipedia: 7 out of 6821

- UnseenMention(wikilinksNED): 114 out of 10000

**Metrics.** Many previous works (De Cao et al., 2021; Ganea and Hofmann, 2017; Ayoola et al., 2022) report the micro-averaged F1 score from Gerbil evaluation tool. In our approach, we always take the top-1 prediction as the final prediction. Since each mention is only associated with one target entity, the micro-averaged F1 score is equivalent to the accuracy in this case (accuracy = precision = recall = F1) We report the accuracy as the evaluation metric for the ED task and didn't use Gerbil evaluation tool.

# E Prompt Construction

In this section, we provide more details on the prompt construction process. The prompt used in our experiments is composed of two parts:

1. *instruction*: a short sentence that describes the task.

2. *demonstration examples*: a set of examples that demonstrate the expected behavior of the model.

We observe that the performance of the model is sensitive to the wording of the instruction and the format of the demonstration examples. To find a good instruction and demonstration examples, we first manually construct a set of instructions and demonstration formats. Then, we randomly sample a few demonstration examples from the training set and manually check whether the model can solve the task with the given instruction and demonstration examples. This process helps us find a good instruction and demonstration format, thought probably not the best. Since our goal is not to find the best instruction and demonstration format, we didn't spend too much time on this process. Below, we provide more details on the prompt construction process for each task.

## E.1 Information Extraction

The instruction used is `Extract the triples in subject-collapsed format from texts below`. The demonstration examples are in the following format: `[input] -> [output]`, where the input

is a text and the output is a set of triples in subject-collapsed format. A concrete example would be:
`Vettaikaaran (2009 film) was originally written in the Tamil language, with B. Babusivan as the screenwriter. -> [s] Vettaikaaran_(2009_film) [r] original language of film or TV show [o] Tamil_language [r] screenwriter [o] B._Babusivan [e]`

## E.2 Entity Disambiguation

We show two prompt construction methods for entity disambiguation.

**Prompt construction A.** The instruction is `Disambiguate the entity surrounded by [START_ENT] and [END_ENT] by giving the correct entity name in the knowledge base`. We randomly select some demonstration examples from the training set. We use the following format to represent the demonstration examples as a string: `[input] -> [mention] [ [target entity] ]`, where the input is a text, the mention is the entity mention surrounded by [START_ENT] and [END_ENT], and the target entity is the entity name in the knowledge base. A concrete example of demo example representation would be: `Eu rejects [START_ENT] German [END_ENT] call to boycott British lamb Peter Blackburn Brussels 1996 08-> German [ Germany ]` The final prompt is a concatenation of the instruction and the demonstration examples. A full prompt with 2 demo examples would be:
`Disambiguate the entity surrounded by [START_ENT] and [END_ENT] by giving the correct entity name in the knowledge base: "Eu rejects [START_ENT] German [END_ENT] call to boycott British lamb Peter Blackburn Brussels 1996 08" -> German [ Germany ]; 16 other items that were put up for auction by [START_ENT] Hendrix [END_ENT] s former girlfriend Kathy Etchingham -> Hendrix [ Jimi Hendrix ]; lead with a well struck header in the seventh minute [START_ENT] Japan [END_ENT] then laid siege to the Syrian penalty area for most -> "`

**Prompt construction B.** The instruction is:
`Disambiguate the entity surrounded by [START_ENT] and [END_ENT] by giving the canonical entity name in the knowledge base:`. The demonstration examples are in the

following format: **[input] -> [mention] [ [target entity] ]**, where the input is a text, the mention is the entity mention surrounded by [START_ENT] and [END_ENT], and the target entity is the canonical entity name in the knowledge base. A concrete example of demo example would be: **"Eu rejects [START_ENT] German [END_ENT] call to boycott British lamb Peter Blackburn Brussels 1996 08 -> German : Canonical form [ Germany ]** A full prompt with 2 demo examples would be: **Disambiguate the entity surrounded by [START_ENT] and [END_ENT] by giving the canonical entity name in the knowledge base: "Eu rejects [START_ENT] German [END_ENT] call to boycott British lamb Peter Blackburn Brussels 1996 08 -> German : Canonical form [ Germany ] 16 other items that were put up for auction by [START_ENT] Hendrix [END_ENT] s former girlfriend Kathy Etchingham -> Hendrix : Canonical form [ Jimi Hendrix ] lead with a well struck header in the seventh minute [START_ENT] Japan [END_ENT] then laid siege to the Syrian penalty area for most -> "**

**Prompt used in each dataset**  We compare the performance of the two prompts in each dataset over the validation set. We select the prompt that performs the best in each dataset. The prompt used in each dataset is shown in Table 6.

| Dataset | Prompt |
|---------|--------|
| AIDA | Prompt 1 |
| MSNBC | Prompt 2 |
| AQUAINT | Prompt 2 |
| ACE2004 | Prompt 2 |
| CWEB | Prompt 2 |
| WIKI | Prompt 1 |

Table 6: Prompt used in each dataset

### E.3   Constituency Parsing

The instruction is: **Perform constituency parsing on the provided sentences in accordance with the Penn TreeBank annotation guidelines.**. The demonstration examples are in the following format: **[input] -> [output]**, where the input is a text and the output is the flattened constituency parse tree. A concrete example would be:

**The dollar weakened against most other major currencies -> [ ( S ( NP-SBJ ( DT The ) ( NN dollar ) ) ( VP ( VBD weakened ) ( PP ( IN against ) ( NP ( RBS most ) ( JJ other ) ( JJ major ) ( NNS currencies ) ) ) ) ) ]**, where the corresponding parse tree is shown in Listing 6.

Listing 6: parse tree with hierarchical indentation
```
( S
    ( NP–SBJ
        ( DT The )
        ( NN dollar )
    )
    ( VP
        ( VBD weakened )
        ( PP
            ( IN against )
            ( NP
                ( RBS most )
                ( JJ other )
                ( JJ major )
                ( NNS currencies )
            )
        )
    )
)
```

One variant of the prompt is to provide a reasoning chain with the demonstration examples. A concrete example would be:
**"The constituency parse tree is: ( S ( NP-SBJ ( NN Bond ) ( NNS prices ) ) ( VP ( VBD were ) ( ADJP-PRD ( RB barely ) ( JJR higher ) ) ) )" "S: This stands for Sentence, the top-level structure in the parse tree." "NP-SBJ: This is the subject noun phrase of the sentence, which is 'Bond prices'." "NN: This stands for Noun, Singular or Mass. In this case, the word 'Bond' falls into this category." "NNS: This stands for Noun, Plural. The word 'prices' is an example of this." "VP: This stands for Verb Phrase, which in this case is 'were barely higher'." "VBD: This stands for Verb, Past Tense. The word 'were' falls into this category." "ADJP-PRD: This stands for Adjective Phrase, used as a predicate. The phrase 'barely higher' is an example of this." "RB: This stands for Adverb. The word 'barely' falls into this category." "JJR: This stands for Adjective, Comparative. The word 'higher' falls into this category.'"**

| | SynthIE-text | | |
| Method | Precision | Recall | F1 |
| --- | --- | --- | --- |
| ***Few-shot Unconstrained*** | | | |
| LLaMA-7B-sc (4 shots) | 9.7 | 8.6 | 9.2 |
| LLaMA-13B-sc (4 shots) | 6.7 | 12.5 | 8.7 |
| LLaMA-33B-sc (4 shots) | 10.6 | 20.6 | 14.0 |
| ***Few-shot Constrained*** | | | |
| LLaMA-7B-sc (4 shots) | 24.1 | 19.3 | 21.5 |
| LLaMA-13B-sc (4 shots) | 26.4 | 30.5 | 28.3 |
| LLaMA-33B-sc (4 shots) | 31.3 | 36.2 | 33.6 |

Table 7: **Results for cIE with subject-collapsed linearization.**

| | SynthIE-text | | |
| Method | Precision | Recall | F1 |
| --- | --- | --- | --- |
| ***Supervised*** | | | |
| GenIE T5-base-fe (Josifoski et al., 2022) | **49.1** | 26.7 | **34.6** |
| ***Few-shot Unconstrained*** | | | |
| LLaMA-7B-feR (4 shots) | 9.9 | 13.5 | 11.4 |
| LLaMA-13B-feR (4 shots) | 10.8 | 17.5 | 13.4 |
| LLaMA-33B-feR (4 shots) | 14.0 | 22.5 | 17.3 |
| Vicuna-13B-feR (4 shots) | 12.5 | 16.7 | 14.3 |
| ***Few-shot Constrained*** | | | |
| LLaMA-7B-feR (4 shots) | 28.1 | 23.6 | 25.7 |
| LLaMA-13B-feR (4 shots) | 31.8 | 31.2 | 31.5 |
| LLaMA-33B-feR (4 shots) | 36.4 | 34.8 | 35.6 |
| Vicuna-13B-feR (4 shots) | 40.3 | 23.8 | 29.9 |

Table 8: **Results for cIE with all relations added to the prompt.** feR stands for fully-expanded linearization with relations added to the prompt.

## F Decoding Settings

We employ constrained beam search during our experiments, with a beam size of 2 and length penalty of 1.0. Our choice of a small beam size is based on our observation that larger beam sizes do not yield significant improvement and may even result in decreased performance. In contrast, a beam size of 2 proves to be sufficient for achieving good results while also being computationally efficient. When evaluating the generated outputs, we select the most probable non-empty generation as the output.

## G Additional Experimental Results

### G.1 Information Extraction

**Subject-Collapsed Linearization.** Subject-collapsed linearization is a variant of linearization where the subject is collapsed into the object. It has the advantage of being more compact (token efficient) than the fully-expanded linearization, but it yields slightly lower performance, intuitively because it's less explicit. Table 7 shows that the constrained decoding approach brings a significant

improvement over the unconstrained decoding approach for LMs of all sizes. Compared with the results in Table 7, the performance of the subject-collapsed linearization is indeed lower than the fully-expanded linearization.

**Adding all relations to the prompt.** With a maximum context length of 2048 tokens, it's not possible to add all the millions of **entities** to the prompt. However, we can add all the **relations** to the prompt, which is a much smaller set (888 relations). One may wonder if adding all the **relations** to the prompt would help the model to learn the relation extraction task better. Because the model can copy the **relation** from the prompt and this may make the task easier. However, we find that adding all the **relations** to the prompt does not help the model to learn the relation extraction task better. Table 8 shows that adding all the **relations** to the prompt only brings a small improvement over the baseline.

**Constituency Parsing with Vicuna.** We also eval-

| Method | Precision | Recall | Tag Accuracy | Validity |
| --- | --- | --- | --- | --- |
| ***Unconstrained*** | | | | |
| Vicuna-7B | 13.5 | 12.7 | 27.8 | 42.2 |
| Vicuna-13B | 31.6 | 28.1 | 29.6 | 51.4 |
| ***Constrained IIG*** | | | | |
| Vicuna-7B | 17.4 | 16.3 | 30.5 | 54.3 |
| Vicuna-13B | 30.8 | 27.7 | 33.4 | 56.1 |
| ***Constrained IDG*** | | | | |
| Vicuna-7B | 35.6 | 31.9 | 41.4 | 100.0 |
| Vicuna-13B | 51.6 | 44.4 | 42.4 | 100.0 |

Table 9: Constituency parsing results with Vicuna. The experiments setting is the same as in Table 3.

uate the constrained decoding approach on the constituency parsing task with the instruction-tuned model Vicuna. Table 9 shows Vicuna's performance is even worse than the LLaMA model.

## H Latency

Here's a concise table comparing the latency of pure decoding to the additional delay introduced by grammar constraints.

**Latency for IE.** We provide a measurement of latency for IE task in Figure 3 and Figure 4. We consider two large grammars, the WikiNER grammar and the REBEL grammar. The WikiNER grammar contains 279 K entities and 158 relations. The REBEL grammar contains 5.9 M entities and 857 relations. Given the grammar, we randomly pick the next token from the set of next allowed tokens

with the end-of-sentence token excluded to avoid early termination. The incremental parsing step is done on CPU and we do not use any optimization techniques such as multi-threading.

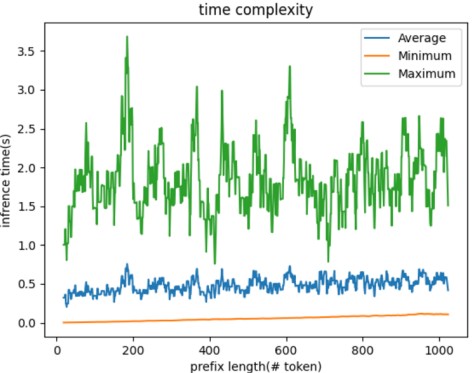

Figure 3: latency of WikiNER grammar

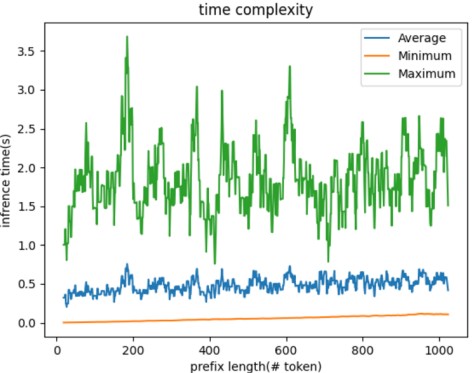

Figure 4: latency of REBEL grammar

As shown in Figure 3s, the latency for WikiNER grammar (0.05s) is comparable to the latency of the LM on GPU. However, the latency for the REBEL grammar (0.5s) is significantly higher than the latency of the LM. We believe this can be largely improved by using more appropriate incremental parser and leave it as future work.

## I    Error Examples of Constituency Parsing

Here we give examples of using LMs to perform CP in a **free-form** generation setting. We show some output examples[1] of LLaMA-13B and Vicuna-13B on CP on Penn Treebank (PTB). We see that while instruction-tuned LMs (Vicuna-13B) can generate seemingly reasonable parse trees, most of them are not correct. On the other hand, base LMs (LLaMA-13B) almost completely fail to follow

the instruction. The generation on the left is from Vicuna-13B and the generation on the right is from LLaMA-13B. The majority of the erroneous output sequences contain unbalanced brackets and length mismatch(missing words or extra words from the input sentence).

---

[1] The visualisation are made from https://chat.lmsys.org/

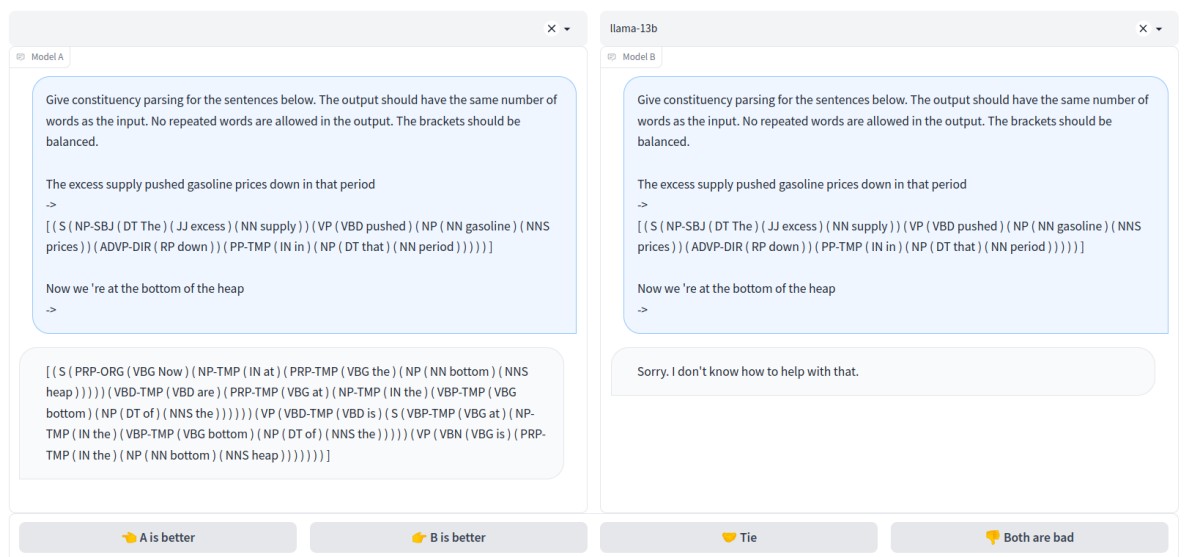

Figure 5: **Example of 1 shot CP on PTB instance No.12** The golden parse tree is *"( S ( ADVP-TMP ( RB Now ) ) ( NP-SBJ ( PRP we ) ) ( VP ( VBP 're ) ( PP-LOC-PRD ( IN at ) ( NP ( NP ( DT the ) ( NN bottom ) ) ( PP ( IN of ) ( NP ( DT the ) ( NN heap ) ) ) ) ) ) )"* The generation from Vicuna-13B is not correct, but it still looks like a reasonable parse tree. The generation from LLaMA-13B fails to follow the instruction.

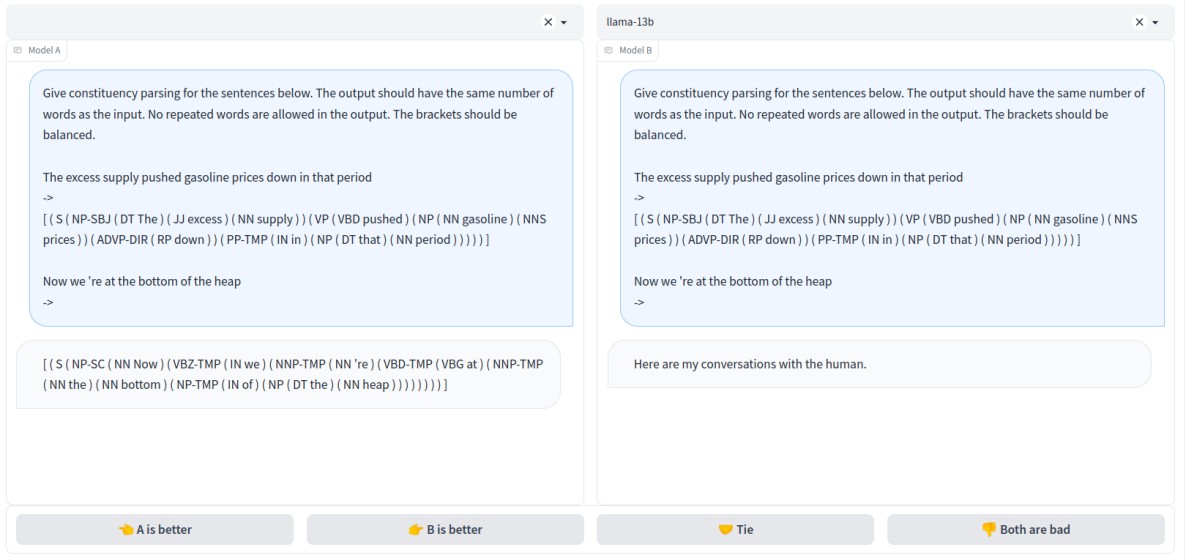

Figure 6: **Example of 1 shot CP on PTB instance No.12** The golden parse tree is *"( S ( ADVP-TMP ( RB Now ) ) ( NP-SBJ ( PRP we ) ) ( VP ( VBP 're ) ( PP-LOC-PRD ( IN at ) ( NP ( NP ( DT the ) ( NN bottom ) ) ( PP ( IN of ) ( NP ( DT the ) ( NN heap ) ) ) ) ) ) )"* The generation from Vicuna-13B looks reasonable, but its bracketing is actually unbalanced. The generation from LLaMA-13B fails to follow the instruction.

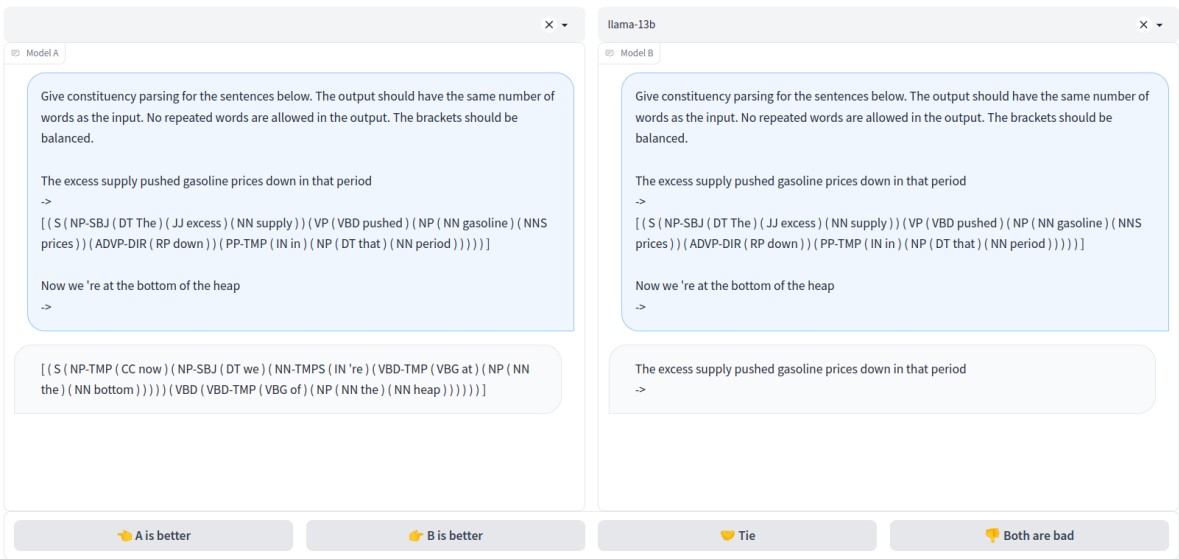

Figure 7: **Example of 1 shot CP on PTB instance No.12** The golden parse tree is *"( S ( ADVP-TMP ( RB Now ) ) ( NP-SBJ ( PRP we ) ) ( VP ( VBP 're ) ( PP-LOC-PRD ( IN at ) ( NP ( NP ( DT the ) ( NN bottom ) ) ( PP ( IN of ) ( NP ( DT the ) ( NN heap ) ) ) ) ) ) )"* The generation from Vicuna-13B is a valid tree structure, but it is not the same as the golden parse tree. The generation from LLaMA-13B fails to follow the instruction.

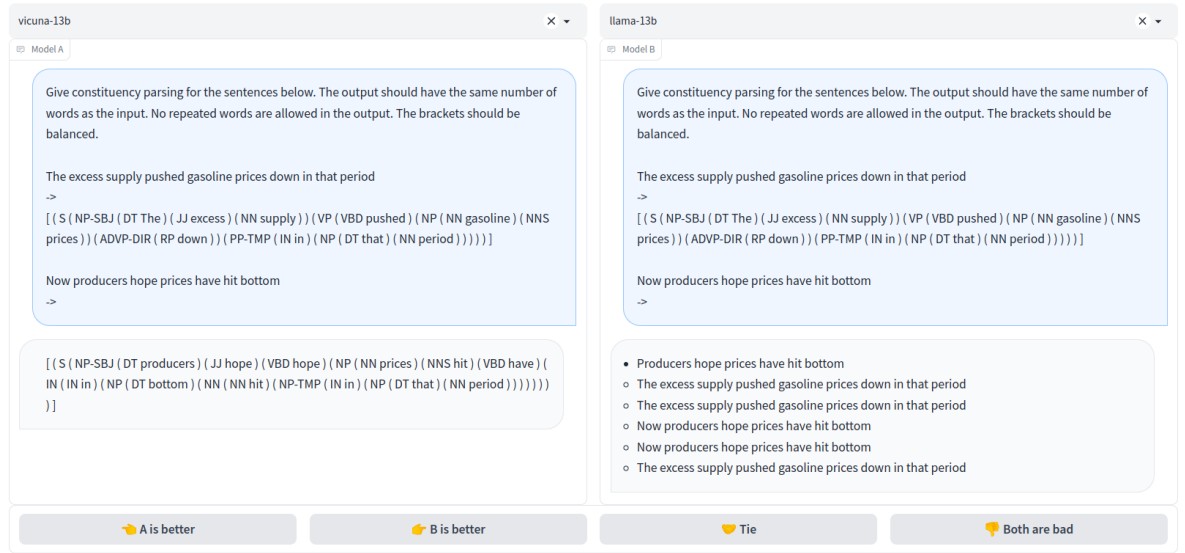

Figure 8: **Example of 1 shot CP on PTB instance No.29** The golden parse tree is *"( S ( ADVP-TMP ( RB Now ) ) ( NP-SBJ ( NNS producers ) ) ( VP ( VBP hope ) ( SBAR ( S ( NP-SBJ ( NNS prices ) ) ( VP ( VBP have ) ( VP ( VBN hit ) ( NP ( NN bottom ) ) ) ) ) ) ) )"* The generation from Vicuna-13B has wrong words(some missing, some extra) and wrong bracketing. The generation from LLaMA-13B fails to follow the instruction.

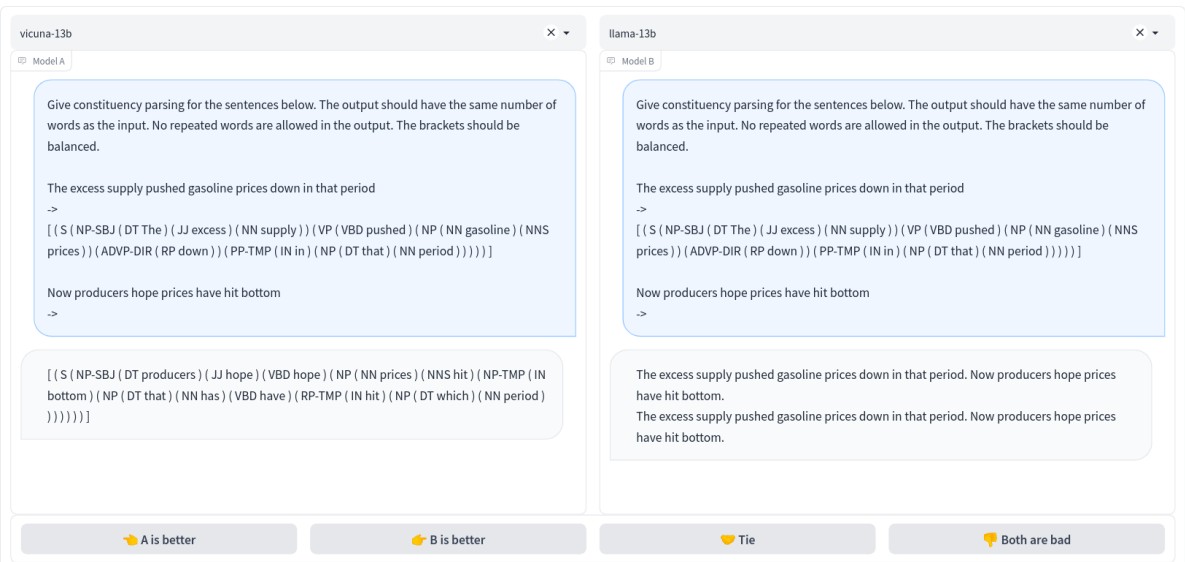

Figure 9: **Example of 1 shot CP on PTB instance No.29 from another sampling** The golden parse tree is *"( S ( ADVP-TMP ( RB Now ) ) ( NP-SBJ ( NNS producers ) ) ( VP ( VBP hope ) ( SBAR ( S ( NP-SBJ ( NNS prices ) ) ( VP ( VBP have ) ( VP ( VBN hit ) ( NP ( NN bottom ) ) ) ) ) ) ) )"* The generation from Vicuna-13B has wrong words(some missing, some extra) and wrong bracketing. The generation from LLaMA-13B fails to follow the instruction.

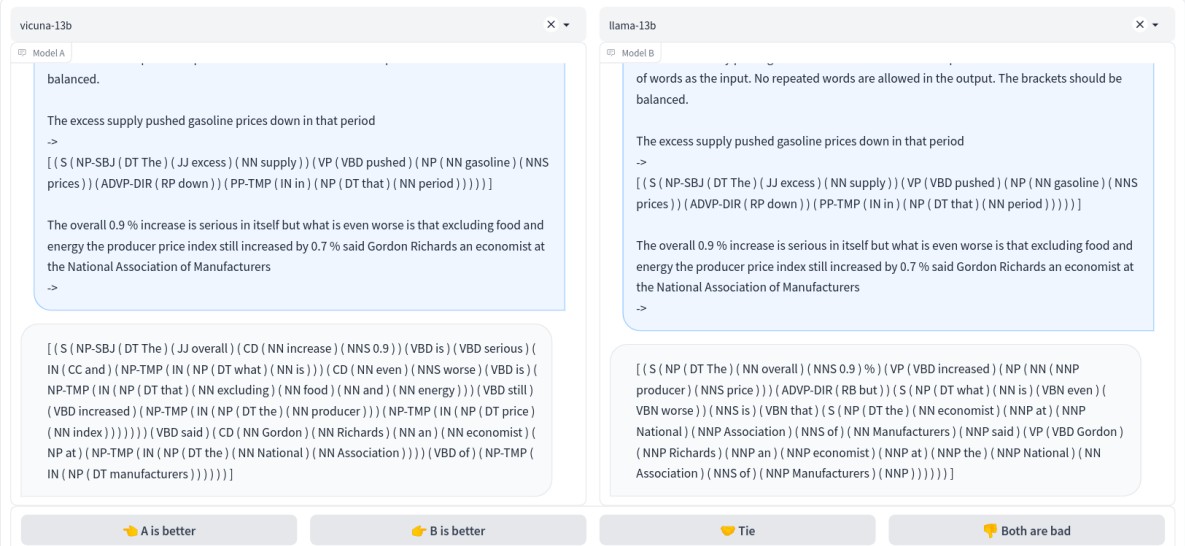

Figure 10: **Example of 1 shot CP on PTB instance No.86(long sentence)** This is a long sentence. The golden parse tree is *"( SINV ( S-TPC-2 ( S ( NP-SBJ ( DT The ) ( JJ overall ) ( ADJP ( CD 0.9 ) ( NN % ) ) ( NN increase ) ) ( VP ( VBZ is ) ( ADJP-PRD ( JJ serious ) ) ( PP ( IN in ) ( NP ( PRP itself ) ) ) ) ) ( CC but ) ( S ( SBAR-NOM-SBJ ( WHNP-1 ( WP what ) ) ( S ( VP ( VBZ is ) ( ADJP-PRD ( RB even ) ( JJR worse ) ) ) ) ) ( VP ( VBZ is ) ( SBAR-PRD ( IN that ) ( S ( PP ( VBG excluding ) ( NP ( NN food ) ( CC and ) ( NN energy ) ) ) ( NP-SBJ ( DT the ) ( NN producer ) ( NN price ) ( NN index ) ) ( ADVP-TMP ( RB still ) ) ( VP ( VBD increased ) ( PP-EXT ( IN by ) ( NP ( CD 0.7 ) ( NN % ) ) ) ) ) ) ) ) ) ) ( VP ( VBD said ) ) ( NP-SBJ ( NP ( NNP Gordon ) ( NNP Richards ) ) ( NP ( NP ( DT an ) ( NN economist ) ) ( PP-LOC ( IN at ) ( NP ( NP ( DT the ) ( NNP National ) ( NNP Association ) ) ( PP ( IN of ) ( NP ( NNP Manufacturers ) ) ) ) ) ) ) )"* The generation from Vicuna-13B has wrong words(some missing, some extra) and wrong bracketing. The generation from LLaMA-13B is not a valid parse tree, e.g. the last constituent *NNP* doesn't have a corresponding word.