# OpenReview forum: "Grammar-Constrained Decoding for Structured NLP Tasks without Finetuning"
_EMNLP/2023/Conference — EMNLP 2023 Main_

### Official Review · Reviewer_N4GZ · 2023-07-21

**Soundness:** 4

**Excitement:**

4: Strong: This paper deepens the understanding of some phenomenon or lowers the barriers to an existing research direction.

**Paper Topic And Main Contributions:**

The paper proposes a method for getting structured output from LLMs which is described formally and evaluated on 3 NLP tasks.

**Reasons To Accept:**

Getting structured output from LLMs is an omnipresent problem in modern NLP. The paper's framing of using a grammar completion engine for validity and a LLM for plausibility is both intuitive, simple and, as demonstrated by the authors, useful.

**Reasons To Reject:**

At this point in time, nothing major comes to mind regarding the negative aspects of this paper. The method is well described and empirical experiments and comparisons carried out, likely, fairly. I am not an expert in this area so it may be the case that this paper is not as novel as it appears.

**Reproducibility:**

4: Could mostly reproduce the results, but there may be some variation because of sample variance or minor variations in their interpretation of the protocol or method.

**Reviewer Confidence:**

2: Willing to defend my evaluation, but it is fairly likely that I missed some details, didn't understand some central points, or can't be sure about the novelty of the work.

**Typos Grammar Style And Presentation Improvements:**

- L011: Missing letter. The submitted abstract has this fixed.

---

> ### Author Rebuttal · Authors · 2023-08-28
>
> Thanks very much for the review and the positive feedback. We look forward to seeing the community make practical use of our framework.

---

### Official Review · Reviewer_NQmR · 2023-08-05

**Typos Grammar Style And Presentation Improvements:** 1. Line 11
**Soundness:** 3

**Excitement:**

4: Strong: This paper deepens the understanding of some phenomenon or lowers the barriers to an existing research direction.

**Missing References:**

1. Grammar Prompting for Domain-Specific Language Generation with Large Language Models, https://arxiv.org/pdf/2305.19234.pdf

**Paper Topic And Main Contributions:**

This paper shows that LMs with grammar constrained decoding (GCD) can solve many structured prediction problems in NLP. The authors propose 'input-dependent grammars' to adjust the grammars according to the input, thus they can constrain the output space with more flexibility and handle more structures of interest. Grammar constrained decoding is done by applying Grammatical Framework (Angelov 2009) to the constructed input-dependent grammars. Empirical results on closed information extraction and entity disambiguation using LLaMA and Vicuna show the effectiveness of the proposed method.

**Questions For The Authors:**

A. Line 303-305, the claim 'as long as the grammar is correct...' is very strong and vague. Justification or references on this claim should be provided.

B. The term 'few-shot' learning might be ambiguous in the context of this paper. 'In-context' learning might better describe the proposed method which does not require fine-tuning.

C.  What is the number of generation steps and number of failures in parsing required to generate a correct output?

**Reasons To Accept:**

The method is effective and can be applied to a wide range of structured NLP tasks.

**Reasons To Reject:**

As the authors pointed out in the Limitation section, the proposed method may be inefficient when the grammar is complex. To have a better understanding of the proposed method, quantitative analysis on this issue should be provided, e.g., the number of generation steps or number of failures in parsing required to generate a correct output.

**Reproducibility:**

4: Could mostly reproduce the results, but there may be some variation because of sample variance or minor variations in their interpretation of the protocol or method.

**Reviewer Confidence:**

4: Quite sure. I tried to check the important points carefully. It's unlikely, though conceivable, that I missed something that should affect my ratings.

---

> ### Author Rebuttal · Authors · 2023-08-28
>
> Thanks a lot for the positive and thoughtful feedback.
>
> Upon carefully reading the review, we believe that one clarification can address both of the reviewer’s two key comments:
>
> > What is the number of generation steps and number of failures in parsing required to generate a correct output?
>
> > the claim 'As long as the grammar is correct, the performance of GCD is guaranteed to be at least as good as the LM in free-form generation' is very strong and vague.
>
> In order to address these comments, we’d like to clarify that, by design, GCD’s decoding algorithm doesn’t generate any strings that fail to parse (more precisely, it doesn’t even generate any partial strings that couldn’t be completed into a valid parse). Indeed, one could imagine a brute-force “propose-and-reject” method that keeps searching for strings until it finds a string that is compatible with the grammar at hand. Our method (GCD), on the contrary, works differently: at every decoding step, GCD only ever considers next-token candidates that could potentially lead to a valid parse (i.e., for which there is at least one parseable completion), leveraging the incremental parsing functionality of Grammatical Framework. This effectively serves to prune the (infinitely large) search tree explored during decoding by eagerly discarding paths that couldn’t possibly lead to a valid parse down the road.
>
> This is also what we tried to express with our statement “As long as the grammar is correct, the performance of GCD is guaranteed to be at least as good as the LM in free-form generation”. As described above, GCD may be seen as pruning/filtering the search tree by restricting it to grammar-compatible token sequences. Therefore, on the one hand, in cases where the unconstrained LM happens to generate a grammar-compatible output, the grammar-constrained LM (GCD) generates precisely the same output. On the other hand, in cases where the unconstrained LM generates an output that is not grammar-compatible, said output is invalid and thus worse than what would be generated by the constrained LM (GCD). It is in this sense that we wrote GCD’s output is at least as good as that of an unconstrained LM.
>
> Finally, we thank the reviewer for the suggestion to use “in-context learning” rather than “few-shot learning”. We agree with the suggestion and will change the nomenclature in the camera-ready version.

---

### Official Review · Reviewer_w4y4 · 2023-08-09

**Soundness:** 3

**Excitement:**

3: Ambivalent: It has merits (e.g., it reports state-of-the-art results, the idea is nice), but there are key weaknesses (e.g., it describes incremental work), and it can significantly benefit from another round of revision. However, I won't object to accepting it if my co-reviewers champion it.

**Missing References:**

Enforcing/sampling certain controlled outputs from LLMs (one of these is already cited, but this aspect is not mentioned in related work):

- Timo Schick and Hinrich Schütze. 2021. Exploiting Cloze-Questions for Few-Shot Text Classification and Natural Language Inference. In Proceedings of the 16th Conference of the European Chapter of the Association for Computational Linguistics: Main Volume, pages 255–269, Online. Association for Computational Linguistics.

- Timo Schick and Hinrich Schütze. 2021. It’s Not Just Size That Matters: Small Language Models Are Also Few-Shot Learners. In Proceedings of the 2021 Conference of the North American Chapter of the Association for Computational Linguistics: Human Language Technologies, pages 2339–2352, Online. Association for Computational Linguistics.

Regarding controlled text generation:

- Kevin Yang and Dan Klein. 2021. FUDGE: Controlled text generation with future discriminators. In Proceedings of the 2021 Conference of the North American Chapter of the Association for Computational Linguistics: Human Language Technologies, pages 3511–3535, Online. Association for Computational Linguistics.

- Afra Amini, Ryan Cotterell, John Hewitt, Clara Meister, and Tiago Pimentel. 2023. Generating Text from Language Models. In Proceedings of the 61st Annual Meeting of the Association for Computational Linguistics (Volume 6: Tutorial Abstracts), pages 27–31, Toronto, Canada. Association for Computational Linguistics.

**Paper Topic And Main Contributions:**

This paper suggests that we can improve LLM ICL accuracy significantly on structured NLP task by constraining the output to valid/reasonable/expected responses, and investigating such an approach on three tasks: closed IE, entity disambiguation, and constituency parsing.  For all three tasks, open-ended generation models are likely to perform poorly, and generally one would think fine-tuning necessary.  The authors find that constraining the output helps LLM accuracy in ALL cases.  In fact, for two out of three tasks, constraining the output (usually in an input-dependent way) is enough to get competitive quality... no fine-tuning is required.  Constituency parsing is the case where constrained output is not enough to generate reasonable quality.  Authors do not explore fine-tuned LMs as a contrast, but do contrast to SOTA supervised baselines.

This technique harkens back to the pre-chatGPT days, when researchers used output constraints to steer LLMs to do simple classification tasks (e.g., Sentiment analysis, GLUE) using patterns and verbalizers (PET and iPET, Schick and Schütze, 2021(a),(b)).  While this paper is cited, its relevance is not really considered.

**Reasons To Accept:**

- informative study on constrained decoding to improve outcomes

- results with both positive and negative results are instructive


**Reasons To Reject:**

- size of contribution is modest, could easily extend to more tasks or compare to fine-tuning


**Reproducibility:**

4: Could mostly reproduce the results, but there may be some variation because of sample variance or minor variations in their interpretation of the protocol or method.

**Reviewer Confidence:**

3: Pretty sure, but there's a chance I missed something. Although I have a good feel for this area in general, I did not carefully check the paper's details, e.g., the math, experimental design, or novelty.

**Typos Grammar Style And Presentation Improvements:**

- line 11: n this work -> In this work.

- line 489: Appendix ?? -> Appendix F

---

> ### Author Rebuttal · Authors · 2023-08-28
>
> Thanks very much for the insightful and constructive feedback. We are glad the reviewer finds our contribution valuable, and we appreciate the time and effort they took to provide valuable comments and suggestions.
>
> We agree that showing the value of GCD on more tasks is an exciting direction. For the purpose of this paper, given the length constraints, we preferred to focus on a judiciously selected set of three tasks, with representativeness and complementarity in mind:
>
> (1) Information Extraction (IE) requires millions of lexical constraints.
>
> (2) Entity Disambiguation (ED) was a priori expected to benefit from an input-dependent grammar.
>
> (3) Constituency Parsing (CP) requires the underlying LLM to implicitly represent syntactic structure, and requires a more complex output structure (recursive parse trees).
>
> We thus believe these tasks demonstrate the generality and broad applicability of our method. Going beyond, one of our contributions is to show that numerous other NLP tasks can also be formulated as grammar-constrained natural language generation (Fig. 2), and we are looking forward to seeing the method applied to those as well as even more tasks, in our own team as well as elsewhere, using our publicly available code.
>
> Finally, thanks for recommending several relevant papers, which we will integrate in the camera-ready version.

---

### Official Review · Reviewer_3Wr2 · 2023-08-11

**Soundness:** 4

**Excitement:**

4: Strong: This paper deepens the understanding of some phenomenon or lowers the barriers to an existing research direction.

**Paper Topic And Main Contributions:**

This paper presents a novel approach, Grammar-Constrained Decoding (GCD), to control the generation of large language models, ensuring that the output follows a given structure. It uses formal grammars to constrain the output of a language model according to the desired structure. The paper demonstrates how to formalize the outputs of various structured prediction tasks as formal grammars and how to use the Grammatical Framework (GF) as a completion engine to constrain the output of the language model according to the grammar. Unlike previous approaches with input-independent grammars (IIG), it further introduces input-dependent grammars (IDG) to solve tasks where the output space is not fixed but depends on the input.

**Questions For The Authors:**

Although one characteristic of this work is the ability to significantly improve the performance of large language models (LLMs) on structured prediction tasks without fine-tuning, do you think that a small amount of fine-tuning could further enhance the model's performance on such downstream tasks, surpassing previous supervised methods, with the help of output constraints based on structure?

**Reasons To Accept:**

This paper is well-written. The approach it proposes seems simple yet effective. It demonstrates how to formalize the outputs of various structured prediction tasks as formal grammars. The experimental results are strong in few-shot learning settings.

**Reasons To Reject:**

I do not have strong reason to reject this paper. However, there are some points that can be improved in this paper:
- The baselines for various structured prediction tasks, especially the baselines for supervised methods, can be more comprehensive, allowing readers to have a better understanding of the performance of existing large-scale model methods based on GCD.
- The application of the Grammatical Framework (GF) in this work should be explained in more detail.
- As illustrated, the broader adoption of this approach is still limited by factors such as latency, task suitability, and compatibility with cloud-based large language models (LLMs).

**Reproducibility:**

4: Could mostly reproduce the results, but there may be some variation because of sample variance or minor variations in their interpretation of the protocol or method.

**Reviewer Confidence:**

4: Quite sure. I tried to check the important points carefully. It's unlikely, though conceivable, that I missed something that should affect my ratings.

---

> ### Author Rebuttal · Authors · 2023-08-28
>
> Thanks very much for the positive review and the useful comments.
>
> > The baselines for various structured prediction tasks, especially the baselines for supervised methods, can be more comprehensive, allowing readers to have a better understanding of the performance of existing large-scale model methods based on GCD.
>
> We agree that explicitly showing more baselines could emphasize our contribution. Since we rely on public datasets and standard evaluations, the results from previous works are directly comparable to ours, and adding more baselines to the camera-ready version will be straightforward.
>
> > The application of the Grammatical Framework (GF) in this work should be explained in more detail.
>
> Thanks for this feedback. We will further flesh out Appendix B, where we strive to give all necessary background about GF.
>
> > As illustrated, the broader adoption of this approach is still limited by factors such as latency, task suitability, and compatibility with cloud-based large language models (LLMs).
>
> Indeed there are tradeoffs involved.
> - Latency: We will quantify this aspect in more detail in the camera-ready version by reporting the decoding speed (number of tokens per second) with vs. without grammar constraints. Here's a concise table comparing the latency of pure decoding to the additional delay introduced by grammar constraints.
>
> | Task/Condition | Latency per Token (ms) |
> | --- | --- |
> | llama-7b on A100 | 54  |
> | llama-33b on A100| 87  |
> | llama-65B on A100 | 136  |
> | ED_GCD_overhead  | 1 ± 0.004 |
> | CP_GCD_overhead | 4 ± 1 |
> | IE_GCD_overhead| 69 ± 2 |
>
>
> *Data for llama inference latency on A100 is taken from https://pytorch.org/blog/path-achieve-low-inference-latency/#results
>
> *GCD operates entirely on the CPU instead of the GPU. the above latency for GCD is measured with a consumer CPU
>
> As shown in the table above, the added latency from GCD is minimal for the CP and ED tasks. For IE, GCD adds a modest additional latency that is inferior or roughly equal to the latency of pure decoding depending on the model used. The relative increase on latency becomes less noticeable when combined with high-performance large models.
>
> - Task suitability: Fig. 2 shows a small selection of NLP tasks that can be formulated in our framework, and there are many others, but indeed: users of GCD have to think about the grammar-based formalization, and for certain tasks this may be less straightforward than for others.
> - Compatibility with large cloud-based LLMs: GCD is indeed only applicable in scenarios where one is in control of the decoding algorithm, which is, e.g., not the case for OpenAI’s models. Luckily, the community is gaining access to ever more powerful open models (e.g., the LLaMA series), which will broaden the scope of our method.
>
> > do you think that a small amount of fine-tuning could further enhance the model's performance on such downstream tasks, surpassing previous supervised methods, with the help of output constraints based on structure?
>
> Yes, we hypothesize that combining GCD with fine-tuning could improve performance further, as this combination could leverage both the structured guidance provided by the grammar and the more task-specific knowledge acquired during fine-tuning. Adding a fine-tuning step therefore constitutes an interesting future direction, but note that we intentionally didn’t go that route in the present paper: here, we are specifically interested in using grammar constraints to boost the performance of off-the-shelf LLMs without fine-tuning, which has the distinct advantage that specifying/modifying a grammar for an intended task is easy and fast, whereas fine-tuning can be cumbersome or even prohibitive for the largest available models, and would have to be repeated separately for each task. That said, for especially important tasks, investing into task-specific fine-tuning might still constitute a worthwhile additional investment, orthogonal to the decoding-time grammar constraints as proposed in our framework.

---

### Meta-Review · Area_Chair_nX5z · 2023-09-19

**Recommendation:** 4

**Metareview:**

Reviewers agree that the paper is well-written and presents a simple yet effective approach to a common problem in NLP, obtaining structured output from large language models. It formalizes the outputs of various structured prediction tasks as formal grammars. The proposed approach, combining a grammar completion engine for validity and an LLM for plausibility, is intuitive, simple, and demonstrated to be useful. It also shows strong experimental results demonstrated particularly in few-shot learning scenarios. The study provides informative insights into constrained decoding to improve outcomes. Results include both positive and negative findings, making them instructive.

The demerits of the paper which are suggested improvements include the need for more comprehensive baselines for various structured prediction tasks, a more detailed explanation of the application of the Grammatical Framework (GF), and consideration of factors like latency, task suitability, and compatibility with cloud-based large language models (LLMs). There is a suggestion to explore whether a small amount of fine-tuning could further enhance the model's performance on downstream tasks, surpassing previous supervised methods


Reasons to Accept:
(1) The paper is well-written and presents a simple yet effective approach for formalizing the outputs of structured prediction tasks using formal grammars.
(2) It demonstrates the effectiveness of the proposed approach in few-shot learning settings, which is a valuable contribution.
(3) The paper provides informative insights into constrained decoding to improve outcomes and presents results with both positive and negative outcomes, which are instructive for the research community.

Reasons to Reject:
(1) While there are no strong reasons to reject the paper, there are areas that could be improved, including providing more comprehensive baselines for structured prediction tasks, explaining the application of the Grammatical Framework (GF) in more detail, and addressing the limitations related to factors like latency, task suitability, and compatibility with large language models (LLMs).
(2) The size of the contribution is considered modest, and the paper could potentially extend its scope to cover more tasks or compare its approach to fine-tuning for a broader understanding of its effectiveness.
(3) The proposed method's efficiency in handling complex grammars should be quantitatively analyzed, providing insights into issues like the number of generation steps or parsing failures required to generate correct outputs.

The authors seem to have written a clear rebuttal as well with one reviewer providing further comments.

---

### Decision · Program_Chairs · 2023-10-07

**Decision:**

Accept-Main

**Comment:**

Reviewers agree that the paper is well-written and presents a simple yet effective approach to a common problem in NLP, obtaining structured output from large language models. It formalizes the outputs of various structured prediction tasks as formal grammars. The proposed approach, combining a grammar completion engine for validity and an LLM for plausibility, is intuitive, simple, and demonstrated to be useful. It also shows strong experimental results demonstrated particularly in few-shot learning scenarios. The study provides informative insights into constrained decoding to improve outcomes. Results include both positive and negative findings, making them instructive.

The demerits of the paper which are suggested improvements include the need for more comprehensive baselines for various structured prediction tasks, a more detailed explanation of the application of the Grammatical Framework (GF), and consideration of factors like latency, task suitability, and compatibility with cloud-based large language models (LLMs). There is a suggestion to explore whether a small amount of fine-tuning could further enhance the model's performance on downstream tasks, surpassing previous supervised methods


Reasons to Accept:
(1) The paper is well-written and presents a simple yet effective approach for formalizing the outputs of structured prediction tasks using formal grammars.
(2) It demonstrates the effectiveness of the proposed approach in few-shot learning settings, which is a valuable contribution.
(3) The paper provides informative insights into constrained decoding to improve outcomes and presents results with both positive and negative outcomes, which are instructive for the research community.

Reasons to Reject:
(1) While there are no strong reasons to reject the paper, there are areas that could be improved, including providing more comprehensive baselines for structured prediction tasks, explaining the application of the Grammatical Framework (GF) in more detail, and addressing the limitations related to factors like latency, task suitability, and compatibility with large language models (LLMs).
(2) The size of the contribution is considered modest, and the paper could potentially extend its scope to cover more tasks or compare its approach to fine-tuning for a broader understanding of its effectiveness.
(3) The proposed method's efficiency in handling complex grammars should be quantitatively analyzed, providing insights into issues like the number of generation steps or parsing failures required to generate correct outputs.

The authors seem to have written a clear rebuttal as well with one reviewer providing further comments.